# Bone microstructure and the evolution of growth patterns in Permo-Triassic therocephalians (Amniota, Therapsida) of South Africa

Adam K. Huttenlocker[1,4] and Jennifer Botha-Brink[2,3]

[1] Department of Biology and Burke Museum of Natural History and Culture, University of Washington, Seattle, Washington, USA
[2] Department of Karoo Palaeontology, National Museum, Bloemfontein, South Africa
[3] Department of Zoology and Entomology, University of the Free State, Bloemfontein, South Africa
[4] Current affiliation: Department of Biology, University of Utah & Natural History Museum of Utah, Salt Lake City, UT, USA

Corresponding author
Adam K. Huttenlocker,
ahuttenlocker@gmail.com

## ABSTRACT

Therocephalians were a speciose clade of nonmammalian therapsids whose ecological diversity and survivorship of the end-Permian mass extinction offer the potential to investigate the evolution of growth patterns across the clade and their underlying influences on post-extinction body size reductions, or 'Lilliput effects'. We present a phylogenetic survey of limb bone histology and growth patterns in therocephalians from the Middle Permian through Middle Triassic of the Karoo Basin, South Africa. Histologic sections were prepared from 80 limb bones representing 11 genera of therocephalians. Histologic indicators of skeletal growth, including cortical vascularity (%$CV$) and mean primary osteon diameters ($POD$), were evaluated in a phylogenetic framework and assessed for correlations with other biologically significant variables (e.g., size and robusticity). Changes in %$CV$ and $POD$ correlated strongly with evolutionary changes in body size (i.e., smaller-bodied descendants tended to have lower %$CV$ than their larger-bodied ancestors across the tree). Bone wall thickness tended to be high in early therocephalians and lower in the gracile-limbed baurioids, but showed no general correlation with cross-sectional area or degree of vascularity (and, thus, growth). Clade-level patterns, however, deviated from previously studied within-lineage patterns. For example, *Moschorhinus*, one of few therapsid genera to have survived the extinction boundary, demonstrated higher %$CV$ in the Triassic than in the Permian despite its smaller size in the extinction aftermath. Results support a synergistic model of size reductions for Triassic therocephalians, influenced both by within-lineage heterochronic shifts in survivor taxa (as reported in *Moschorhinus* and the dicynodont *Lystrosaurus*) and phylogenetically inferred survival of small-bodied taxa that had evolved short growth durations (e.g., baurioids). These findings mirror the multi-causal Lilliput patterns described in marine faunas, but contrast with skeletochronologic studies that suggest slow, prolonged shell secretion over several years in marine benthos. Applications of phylogenetic comparative methods to new histologic data will continue to improve our understanding of the evolutionary dynamics of growth and body size shifts during mass extinctions and recoveries.

## INTRODUCTION

Mass extinctions are frequently followed by short-term reductions in body sizes of survivor lineages, a pattern known as the 'Lilliput effect' (*Urbanek, 1993*; *Harries, Kauffman & Hansen, 1996*). However, in the absence of adequate phylogenetic and life history data, the mechanisms of size reductions can be unclear and likely differ across environments, taxonomic groups, and extinction events (*Twitchett, 2007*; *Harries & Knorr, 2009*). Lilliput patterns have been documented widely in marine invertebrate groups following the end-Permian extinction (*Payne, 2005*; *Twitchett, 2007*; *Luo et al., 2008*; *Mutter & Neuman, 2009*; *Metcalfe, Twitchett & Price-Lloyd, 2011*; *Song, Tong & Chen, 2011*; *Rego et al., 2012*), and anecdotally in tetrapods of the Triassic *Lystrosaurus* Assemblage Zone in the Karoo Basin of South Africa (ca. 252.3 Ma), but growth dynamics underlying these patterns are not fully understood. Therocephalians represent an exemplary clade of nonmammalian therapsids that thrived from the Middle Permian to Middle Triassic, and survived the end-Permian extinction as important components of Triassic survivor and recovery faunas in the Karoo Basin (*Botha & Smith, 2006*). In addition to the dicynodont *Lystrosaurus*, at least three genera of therocephalians in two major groups have observed stratigraphic occurrences that span the extinction boundary in the Karoo: the bauroid *Tetracynodon*, and the akidnognathids *Promoschorhynchus* and *Moschorhinus* (*Smith & Botha, 2005*; *Botha & Smith, 2006*; *Huttenlocker, Sidor & Smith, 2011*). Other Triassic taxa (e.g., *Olivierosuchus*, *Regisaurus*) have long ghost lineages extending into the Permian, indicating that they too survived the extinction but lack a Permian record within the depositional basin (*Huttenlocker, 2009*; *Huttenlocker, Sidor & Smith, 2011*). Although therocephalians are generally exceeded in abundance by dicynodont therapsids in the Karoo Basin, their diversity, extensive stratigraphic range, and success during the end-Permian extinction make them an ideal group to study evolutionary patterns during the Permian-Triassic transition.

Previous morphological studies of therocephalians have emphasized their functional anatomy, including jaw mechanics and locomotory specializations (e.g., *Kemp, 1972a*; *Kemp, 1972b*; *Kemp, 1978*; *Kemp, 1986*; *Fourie & Rubidge, 2007*; *Fourie & Rubidge, 2009*). Recent collaborative work on therocephalians has emphasized integration of their fine structure and internal anatomy to resolve paleobiological questions, including tooth replacement patterns, braincase structure, and growth and histomorphologic structure (*Abdala, Rubidge & van den Heever, 2008*; *Sigurdsen et al., 2012*; *Huttenlocker & Botha-Brink, 2013*). Detailed investigations of histmorphology are particularly useful, permitting assessments of growth patterns and variation within or among closely related therapsid species (e.g., *Botha & Angielczyk, 2007*; *Huttenlocker & Botha-Brink, 2013*). Moreover, as growth patterns are associated with organismal fitness, recent investigations

into bone microstructure have inquired into whether certain growth strategies conferred success on some groups during the end-Permian mass extinction (e.g., rapid growth in the dicynodont *Lystrosaurus* and its relatives; *Botha-Brink & Angielczyk, 2010*).

## Bone microstructure in therapsids

Bone histology has offered insights into the lifestyles and growth patterns of many of the major subclades of nonmammalian therapsids. Recent examples include investigations of feeding and locomotion, habitat use, and especially growth dynamics (e.g., *Ray, Chinsamy & Bandyopadhyay, 2005*; *Jasinoski, Rayfield & Chinsamy, 2010*; *Chinsamy-Turan, 2012*). Earlier surveys of bone histology emphasized differences between basal (pelycosaurian-grade) synapsid and therapsid tissue composition, matrix organization, and degree of vasculature of the limb bones (*Enlow & Brown, 1957*; *Enlow, 1969*; *Ricqlès, 1969*; *Ricqlès, 1974a*; *Ricqlès, 1974b*; *Ricqlès, 1976*). Particularly, fibrolamellar tissue complexes (vascularized bone tissues that incorporate primary osteons within a woven-fibered matrix) were found to be near ubiquitous among limb elements of sampled therapsids, suggesting that this tissue complex appeared early during therapsid evolution prior to the origination of mammals bearing this tissue-type (*Ricqlès, 1969*; *Ricqlès, 1974a*; *Ricqlès, 1974b*; *Ray, Botha & Chinsamy, 2004*; *Chinsamy & Hurum, 2006*; *Ray, Bandyopadhyay & Bhawal, 2009*). Indeed, fibrolamellar bone has been reported in basal therapsids (e.g., *Biarmosuchus*: *Ricqlès, 1974b*), as well as in some immature pelycosaurian-grade synapsids (e.g., *Sphenacodon* and *Dimetrodon* juveniles) and in fast-growing portions of the skeleton (e.g., the elongated neural spines of *Dimetrodon* and crossbars on the neural spines of *Edaphosaurus*) (*Huttenlocker, Rega & Sumida, 2010*; *Huttenlocker, Mazierski & Reisz, 2011*; *Huttenlocker & Rega, 2012*; *Shelton et al., 2013*). However, there is great histovariability in the organization of fibrolamellar bone even within major subclades of therapsids. Fibrolamellar bone may be formed by varying degrees of woven- and parallel-fibered interstitial matrix and incorporates a variety of vascular motifs, and may be zonal (punctuated by cyclic growth marks) or azonal. *Ray, Botha & Chinsamy (2004)* reported the presence of zonal fibrolamellar bone in many Permian and Middle Triassic taxa, but suggested that sustained (non-cyclic) growth patterns might have arisen occasionally in a number of phylogenetically disparate taxa that do not encompass the immediate ancestors of mammals (gorgonopsian *Aelurognathus*; eucynodont *Cynognathus*; and some bidentalian dicynodonts). The abundance of dicynodont fossils in Permian and Triassic rocks and recent advances in their systematic relationships have permitted more detailed comparisons of growth patterns in this diverse subclade (*Chinsamy & Rubidge, 1993*; *Botha, 2003*; *Ray & Chinsamy, 2004*; *Ray, Chinsamy & Bandyopadhyay, 2005*; *Botha & Angielczyk, 2007*; *Ray, Bandyopadhyay & Bhawal, 2009*; *Botha-Brink & Angielczyk, 2010*; *Green, Schweitzer & Lamm, 2010*; *Nasterlack, Canoville & Chinsamy-Turan, 2012*; *Ray, Botha-Brink & Chinsamy-Turan, 2012*). Phylogenetic comparative surveys have revealed patterns of increasing tissue vascularity during the evolutionary history of bidentalian dicynodonts (especially in Triassic forms like *Lystrosaurus*), and  determinate

growth patterns with peripheral rest lines and systematic cortical remodeling in large kannemeyeriiforms (*Botha-Brink & Angielczyk, 2010*; *Green, Schweitzer & Lamm, 2010*; *Ray, Botha-Brink & Chinsamy-Turan, 2012*).

The diversity of growth patterns in other nonmammalian therapsid groups, as well as their phylogenetic and temporal distributions, is incompletely known. A body of literature on nonmammalian cynodont histology has accrued in recent years (e.g., *Ricqlès, 1969*; *Botha & Chinsamy, 2000*; *Botha & Chinsamy, 2004*; *Botha & Chinsamy, 2005*; *Ray, Botha & Chinsamy, 2004*; *Chinsamy & Abdala, 2008*; *Botha-Brink, Abdala & Chinsamy, 2012*), but sampling has been more limited in other theriodonts, such as the gorgonopsians and therocephalians (*Ray, Botha & Chinsamy, 2004*; *Chinsamy-Turan & Ray, 2012*; *Huttenlocker & Botha-Brink, 2013*). *Ricqlès (1969)* suggested differential rates of growth between a basal therocephalian from the Middle Permian of South Africa and the Late Permian whaitsiid '*Notosollasia*' (=*Theriognathus*). Given the comparatively more vascularized cortical bone in the radius of the whaitsiid (1969: plate IV), Ricqlès suggested that therocephalians might have exhibited accelerated growth rates later in their evolutionary history, paralleling the aforementioned temporal pattern of increasing growth rates in some dicynodonts. More recently, *Ray, Botha & Chinsamy (2004)* and *Chinsamy-Turan & Ray (2012)* analyzed additional material from an indeterminate scylacosaurid (erroneously identified as '*Pristerognathus*'), in addition to other therapsid material, and argued for similar 'flexible' growth patterns in gorgonopsians, basal therocephalians, and most early cynodonts. The authors suggested that more rigorous taxonomic sampling would better substantiate parallel trends toward a loss in developmental plasticity and acceleration of growth rates as in dicynodonts. Inadequate sampling of eutherocephalians before and after the end-Permian mass extinction limits our understanding of evolutionary patterns in therapsid histomorphology and skeletal growth during this important geologic transition.

## Present study

Although eutherocephalians have not been sampled histologically for such comparisons, recent revisions to Permo-Triassic boundary-crossing taxa have necessitated cursory descriptions of eutherocephalian histology for its ontogenetic and paleobiological implications (e.g., *Tetracynodon*: *Sigurdsen et al., 2012*; *Moschorhinus*: *Huttenlocker & Botha-Brink, 2013*). During the course of this work, we developed a database of histological data and images with the goal of addressing features of life history evolution in Permian and Triassic eutheriodonts (therocephalians and cynodonts), particularly in the context of the end-Permian extinction. Here, we present a reappraisal of limb bone microstructure in Permian and Triassic therocephalians based on new histologic sampling, and offer a hypothesis of the evolution of their growth patterns. We suggest that therocephalians provide a robust study system for investigating the evolution of growth strategies during the Permian-Triassic transition, and a useful point of comparison and contrast to other groups that lived during this time (e.g., dicynodonts, cynodonts).

## MATERIALS AND METHODS

### Specimen selection and histological processing

Specimens were selected based on completeness and availability for histological processing, but a broad sample of the major representatives of South African therocephalians was desired in order to recognize long-term patterns (if present) or clade specific histomorphology. Specimens that were semi-articulated and included diagnostic cranial material were preferred for accuracy of taxonomic identifications. Some specimens were not diagnosable to genus, but were resolved to their respective higher taxon as in the case of five indeterminate scylacosaurids described here. Scylacosaurids are generally difficult to identify unless a complete and accurate antecanine tooth count can be made, and some authors have suggested that the diversity of scylacosaurids is over-split because variations in tooth count may be ontogenetically variable (e.g., *Abdala, Rubidge & van den Heever, 2008*). The sample therefore included 80 limb elements from 33 individuals in 11 genera: *Lycosuchus*, *Glanosuchus*, *Moschorhinus*, *Olivierosuchus*, *Hofmeyria*, *Mirotenthes*, *Theriognathus*, *Ictidosuchoides*, *Tetracynodon*, *Scaloposaurus*, and *Microgomphodon* (Fig. 1), plus additional scylacosaurid material not diagnosed to genus. Thin-sections were prepared using standard histological techniques modified from *Chinsamy & Raath (1992)* and *Wilson (1994)*. Limb bone midshafts were sampled cross-sectionally and imaged using Nikon Eclipse 50i and LV100 POL petrographic microscopes with a digital image capture system. Histomorphometric variables (discussed below) were measured using NIS-Elements and NIH ImageJ v. 1.42q software (*Rasband, 1997–2012*). *Moschorhinus* histomorphology was excluded from present description as it has been discussed elsewhere (*Huttenlocker & Botha-Brink, 2013*), but data from *Moschorhinus* were included in the quantitative analyses (Table 1).

### Bone tissue typology: definitions and selection of growth proxies

Bone tissue texture exhibits marked variation in therocephalians and other therapsids, varying from highly organized and lamellar to disorganized and woven. Only a few recent studies have integrated qualitative and quantitative assessments of tissue texture and vascular proxies of growth in Permo-Triassic therapsids (*Botha & Chinsamy, 2000*; *Botha & Chinsamy, 2004*; *Botha & Chinsamy, 2005*; *Ray & Chinsamy, 2004*; *Ray, Botha & Chinsamy, 2004*; *Ray, Chinsamy & Bandyopadhyay, 2005*; *Ray, Bandyopadhyay & Appana, 2010*; *Botha-Brink & Angielczyk, 2010*; *Huttenlocker & Botha-Brink, 2013*). Generally, cellular bone forms as osteoblasts become incorporated into the extracelluar (or interstitial) matrix (ECM) forming quiescent osteocytes. The overall bone apposition rate affects the texture of the mineralized ECM, with collagen fibers and crystallites bearing a more lamellar organization under slower growth and a nonlamellar (woven-fibered) texture under faster growth. Parallel-fibered bone, an intermediate tissue-type, can be identified by its 'streaky' appearance under polarized light, with the predominant fiber orientation being parallel to the surface of the bone and forming a woven-basket texture in most cases. In cases in which birefringent properties have been disrupted by diagenetic processes, it is possible to approximate the relative organization of mineralized fibers with reference to the

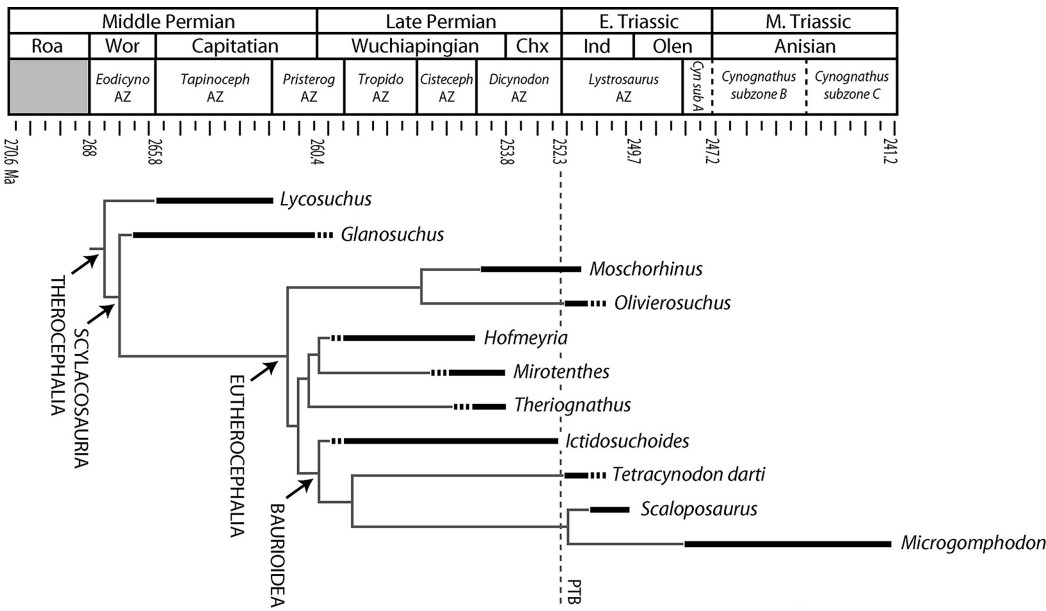

**Figure 1 Stratigraphic ranges of therocephalians sampled histologically in the present study.** Dashed line indicates position of Permian-Triassic Boundary (PTB). Abbreviations: Chx, Changxingian; *Cisteceph AZ*, *Cistecephalus* Assemblage Zone; *Cyn sub A*, *Cynognathus* subzone A; *Eodicyno AZ*, *Eodicynodon* Assemblage Zone; Ind, Induan; Olen, Olenekian; *Pristerog AZ*, *Pristerognathus* Assemblage Zone; Roa, Roadian; *Tapinoceph AZ*, *Tapinocephalus* Assemblage Zone; *Tropido AZ*, *Tropidostoma* Assemblage Zone; Wor, Wordian.

organization of the lacunocanalicular network within the ECM (*Stein & Prondvai, 2013*). In contrast to parallel-fibered bone, woven-fibered bone includes large, globular osteocyte lacunae that are usually densely packed within the mineralized ECM. Nonlamellar tissues (parallel- and woven-fibered) may also frequently incorporate large vascular canals that later become infilled with one or two concentric lamellae forming primary osteons (diagnosed by their 'Maltese cross' pattern of birefringence under polarized light). These osteons house passageways for blood vessels and nerves while also contributing to the structural integrity of the bone by providing added bone mass and helping to blunt microcracks (*Currey, 2002*). The result is a fibrolamellar bone complex (herein 'FLB'), in which a disorganized, fibrous or nonlamellar interstitial matrix incorporates an anastomosing network of centripetally lamellated primary osteons. *Currey (1987)* and *Currey (2002)* defined FLB broadly as a tissue complex formed by parallel- (or woven-) fibered bone with primary osteons (2002: p. 18). By contrast, *Ricqlès (1974a)* originally restricted the term 'fibrolamellar' to tissues formed largely by woven-fibered bone with primary osteons, excluding parallel-fibered bone from this category. Herein, we follow the traditional usage of Ricqlès, but temper this strict definition by noting that parallel- and woven-fibered bone form a continuum that is often ill-defined (and may be present simultaneously in many therapsid bony tissues, even within the same section). Bone cortices formed primarily by lamellar tissue, which forms at relatively slower apposition rates ($\sim$1 μm/day or less) (*de Margerie, Cubo & Castanet, 2002*; *Lee et al., 2013*), do not typically incorporate primary osteons, instead bearing simple vascular canals or being

**Table 1** Specimens, elements, and histometric measurements in studied therocephalians.

| | %Largest[*] | Element | Midshaft cross-sectional area (mm²) | RBT (%) | κ | Cortical vascularity (%) | Mean POD (μm) |
|---|---|---|---|---|---|---|---|
| ***Lycosuchus*** | | | | | | | |
| SAM-PK-9084 | 100% | radius | 296.76 | 16 | 0.42 | 10.7[5.5] | 101[11] |
| | | ulna | 325.40 | 25 | 0.38 | 15.7[2.0] | 109[26] |
| SAM-PK-K9012 | – | femur | 1276.60 | 15 | 0.67 | 16.9[6.2] | 168[17] |
| ***Glanosuchus*** | | | | | | | |
| BP/1/6228 | 47% | ulna | 57.62 | 20 | 0.42 | 04.9[0.9] | 51[13] |
| **Scylacosauridae indet.** | | | | | | | |
| BP/1/5576 | – | radius | 30.99 | 19 | 0.47 | 10.3[2.2] | 68[10] |
| | | ulna | 36.63 | 24 | 0.53 | 09.7[4.3] | 71[08] |
| BP/1/5587 | – | humerus | 169.42 | 33 | 0.29 | 13.3[3.2] | 75[08] |
| | | radius | 88.26 | 23 | 0.46 | 03.5[0.7] | 71[15] |
| | | ulna | 120.25 | 24 | 0.44 | 06.0[0.8] | 85[07] |
| CGS R300 | – | humerus | 308.56 | 31 | 0.32 | 15.9[6.5] | 112[13] |
| SAM-PK-5018 | – | humerus | – | – | – | – | – |
| | | radius | – | 17 | – | – | – |
| | | femur | – | 15 | – | – | – |
| | | tibia | 83.86 | 24 | 0.60 | 10.3[1.5] | 55[10] |
| | | fibula | – | – | – | – | – |
| SAM-PK-11557 | – | fibula | 49.10 | 25 | 0.47 | 07.0[2.8] | 44[07] |
| ***Moschorhinus*** | | | | | | | |
| NMQR 48 | 62% | humerus | 205.85 | 30 | 0.45 | 16.6[5.0] | – |
| | | radius | 48.10 | 26 | 0.40 | 05.2[2.4] | 67[18] |
| | | ulna | 102.67 | 27 | 0.43 | 04.8[1.3] | 67[10] |
| | | femur | 263.13 | 22 | 0.58 | 12.8[4.8] | 95[13] |
| NMQR 3939 | 65% | humerus | 212.25 | 35 | 0.38 | 20.1[6.9] | 134[36] |
| | | radius | 92.77 | 34 | 0.31 | 11.4[2.6] | 101[18] |
| | | ulna | 101.86 | 29 | 0.44 | 10.1[2.1] | 120[27] |
| | | femur | 206.82 | 28 | 0.44 | 18.4[6.9] | 140[32] |
| | | tibia(l) | 103.85 | 37 | 0.30 | 15.5[1.9] | 117[17] |
| | | tibia(r) | 114.28 | 31 | 0.39 | 15.6[5.8] | 119[34] |
| NMQR 1640a | (84%) | femur | 325.70 | 25 | 0.50 | 11.1[4.8] | 96[12] |
| NMQR 1640b | – | tibia | 427.87 | 35 | 0.39 | 09.7[2.5] | 106[23] |
| NMQR 3351 | 91% | femur | 488.65 | 21 | 0.55 | 13.6[5.0] | 100[22] |
| NMQR 3684 | (95%) | femur | 378.12 | 27 | 0.53 | 14.2[4.6] | 92[17] |
| SAM-PK-K118 | 59% | humerus | 161.60 | 28 | 0.41 | 25.5[4.9] | 100[17] |
| | | radius | 62.80 | 23 | 0.54 | 19.5[3.2] | 101[17] |
| SAM-PK-K9953 | 59% | femur | 155.13 | 25 | 0.42 | 20.9[5.8] | 108[15] |
| UCMP 42787 | (67%) | humerus | 209.64 | 35 | 0.34 | 19.5[4.0] | 107[20] |
| | | radius | 83.64 | 35 | 0.27 | 12.4[3.6] | 92[18] |
| | | fibula | 42.53 | 32 | 0.28 | 10.8[1.9] | 58[14] |
| BP/1/4227 | 76% | humerus | 312.00 | 33 | 0.35 | 12.1[2.9] | 100[16] |
Table 1 (*continued*)

| | %Largest[*] | Element | Midshaft cross-sectional area (mm$^2$) | RBT (%) | $\kappa$ | Cortical vascularity (%) | Mean POD (μm) |
|---|---|---|---|---|---|---|---|
| | | radius | 103.60 | 30 | 0.36 | 10.9[2.1] | 88[10] |
| | | ulna | 96.00 | 26 | 0.48 | 08.5[2.0] | 91[19] |
| *Olivierosuchus* | | | | | | | |
| NMQR 3605 | 100% | humerus | 47.54 | 36 | 0.38 | 09.9[1.8] | 99[10] |
| SAM-PK-K10617 | (65%) | femur | 23.72 | 19 | 0.65 | 03.4[1.1] | 75[10] |
| *Hofmeyria* | | | | | | | |
| BP/1/4404 | 69% | humerus | 12.56 | 31 | 0.32 | 07.5[1.4] | 72[11] |
| | | radius (l) | 4.80 | 35 | 0.26 | 03.4[1.0] | 46[05] |
| | | radius (r) | 4.04 | 40 | 0.15 | 04.8[1.9] | 47[05] |
| | | ulna(l) | 3.63 | 39 | 0.22 | 03.0[0.6] | 39[05] |
| | | ulna(r) | 4.30 | 39 | 0.20 | 04.7[1.1] | 43[08] |
| *Mirotenthes* | | | | | | | |
| SAM-PK-K6511 | 78% | humerus | 17.12 | 32 | 0.31 | 05.4[1.5] | 54[07] |
| | | radius | 6.28 | 34 | 0.33 | 03.5[1.5] | 38[07] |
| | | ulna | 5.24 | 32 | 0.42 | 02.0[0.5] | 41[07] |
| | | femur | 19.40 | 25 | 0.42 | 03.6[0.6] | 58[07] |
| | | tibia | 11.28 | 31 | 0.31 | 03.1[0.8] | 38[06] |
| | | fibula | 3.64 | 32 | 0.39 | 03.9[1.1] | 32[04] |
| *Theriognathus* | | | | | | | |
| NMQR 3375 | 38% | femur | 43.45 | 24 | 0.51 | 06.2[2.0] | 65[10] |
| BP/1/719 | (62%) | femur | 140.21 | 20 | 0.59 | 06.8[1.4] | 87[11] |
| *Ictidosuchoides* | | | | | | | |
| SAM-PK-K8659 | 50% | humerus | 23.92 | 22 | 0.56 | 07.3[2.0] | 45[05] |
| | | radius | 10.08 | 24 | 0.52 | 04.2[1.1] | 38[05] |
| | | femur | 45.64 | – | – | – | 38[05] |
| | | tibia | 28.32 | 16 | 0.63 | 02.8[1.0] | 43[08] |
| | | fibula | 10.44 | 22 | 0.59 | 03.8[1.5] | 40[07] |
| SAM-PK-K10423 | (50%) | femur | 34.80 | 14 | 0.71 | 04.7[1.8] | 65[08] |
| | | tibia | 9.52 | 23 | 0.49 | 04.3[1.5] | 39[05] |
| | | fibula | 19.60 | 19 | 0.47 | 04.7[1.6] | 40[08] |
| BP/1/75 | – | humerus | 44.27 | 19 | 0.64 | 11.1[1.4] | 52[06] |
| BP/1/4092 | 100% | humerus | 60.19 | 22 | 0.56 | 10.2[2.1] | 60[07] |
| | | radius | 39.52 | 21 | 0.58 | 11.1[3.1] | 62[08] |
| | | ulna | 74.67 | 26 | 0.54 | 08.3[2.1] | 52[08] |
| *Tetracynodon* | | | | | | | |
| NMQR 3745 | 85% | humerus | 11.44 | 22 | 0.56 | 8.1[3.2] | 36[10] |
| UCMP 78395 | (94%) | humerus | 16.48 | 24 | 0.49 | 4.9[1.7] | 42[05] |
| | | radius | 8.40 | 27 | 0.43 | 1.9[0.5] | 42[10] |
| | | ulna | 6.12 | 31 | 0.33 | 2.8[1.0] | 38[06] |
| | | femur | 18.28 | 17 | 0.66 | 5.0[2.5] | 39[06] |
| | | tibia | 12.48 | 25 | 0.48 | 3.9[0.7] | 42[06] |
| UCMP 78396 | (94%) | humerus | 16.96 | 21 | 0.54 | 4.7[1.9] | 42[12] |

Table 1 (*continued*)

| | %Largest[*] | Element | Midshaft cross-sectional area (mm$^2$) | RBT (%) | κ | Cortical vascularity (%) | Mean POD (μm) |
|---|---|---|---|---|---|---|---|
| | | femur | 18.16 | 17 | 0.62 | 4.3[1.1] | 42[07] |
| | | fibula | 5.19 | 25 | 0.53 | 2.3[0.5] | 35[06] |
| *Scaloposaurus* | | | | | | | |
| SAM-PK-K4638 | 67% | humerus | 15.34 | – | – | 06.6[1.6] | 62[08] |
| *Microgomphodon* | | | | | | | |
| NMQR 3189 | (82%) | humerus | 11.04 | 21 | 0.61 | 08.8[2.1] | 43[08] |
| | | femur | 16.52 | 18 | 0.61 | 06.1[1.6] | 44[08] |
| | | tibia | 7.86 | 24 | 0.51 | 08.0[3.2] | 41[09] |
| | | fibula | 3.33 | 29 | 0.39 | 04.2[0.9] | 32[04] |

**Notes.**

[*] Based on relative basal skull length (BSL). Parentheses indicate estimates for incomplete skulls. Numbers in brackets represent one standard deviation.

avascular. In lamellar bone, the lacunocanalicular network is ordered, the osteocyte lacunae being small and more lenticular in appearance with the long axis oriented parallel to the surface of the bone. Both tissue complexes can be zonal (periodically interrupted by growth marks) or azonal. Growth marks in zonal bone may be present in the form of lines of arrested growth or 'LAGs' (denoted by an opaque cement line, traceable around the entire cortex, and indicative of a temporary cessation of growth) or annuli (thin bands of dense, annular tissue, usually parallel-fibered or lamellar, deposited during periods of slowed growth).

For quantitative histomorphometric analysis, two vascular proxies of skeletal growth were selected: cortical vascularity and mean primary osteon diameter. These proxies were selected in order to evaluate the extent to which histological correlatives of growth varied across phylogeny, and whether their evolution was tied to body size or other biological factors. Vascular proxies have offered useful indicators of skeletal growth in extant and extinct tetrapods (*Castanet et al., 2000*; *de Margerie, Cubo & Castanet, 2002*; *de Margerie et al., 2004*; *Buffrénil, Houssaye & Böhme, 2007*; *Cubo et al., 2012*), and their utility here allows comparisons with other histological studies of therapsids in which similar measures were used (e.g., *Botha-Brink & Angielczyk, 2010*; *Huttenlocker & Botha-Brink, 2013*). Notably, ontogenetic variation in growth may introduce a lesser degree of cortical vascularity in adult bones that exhibited decreasing apposition rates prior to death and burial, and may therefore introduce variation in a single cross-section where this transition is recorded. Nonmammalian therapsids, however, are well suited to relative growth rate estimation based on tissue texture and vascularity, due to their generally thick bone walls (preserving the early record of primary growth) and limited secondary remodeling (*Botha-Brink & Angielczyk, 2010*). Measurement of cortical vascularity (%CV, the relative area of the cortex that is occupied by porous, vascular spaces) follows *Lee et al. (2013)*. Measurements were restricted to the inner two-thirds to three-quarters of the cortex where the bone formed at high, sustained growth rates (*Cubo et al., 2012*), and were averaged from ten quadrants sampled circularly around each midshaft cross-section. The subsampled

quadrants excluded areas of secondary reconstruction and outer regions of simple canals. We also estimated mean primary osteon diameter (*POD*) by measuring in microns the transverse (or minimum) widths of 15 primary osteons visible in the subsampled regions and averaging them across all regions within a given midshaft cross-section.

To examine potential effects of size and robusticity on measured histomorphometric variables, we also estimated two proxies of overall bone robusticity: (1) *K* (the proportional diameter of the medullary region relative to the total diameter of the cross-section; *Currey & Alexander, 1985*) and (2) relative bone wall thickness or '*RBT*' (the percentage of the average cross-sectional thickness of the bone wall relative to the total diameter of the cross-section; *Chinsamy, 1993*). These variables play a frequent role in histological studies of fossil tetrapod bone, as bone robusticity may correspond to habitat preferences or mechanical loading (*Wall, 1983*; *Currey & Alexander, 1985*; *Currey, 2002*; *Laurin, Girondot & Loth, 2004*; *Germain & Laurin, 2005*; *Kriloff et al., 2008*). However, their relationships to size, growth, and vascularity have been underexplored in a comparative framework. These measurements, as well as cross-sectional area at midshaft, were attained using NIH ImageJ, and were tested for correlations with vascular growth proxies using Pearson's product-moment correlation tests. Histomorphometric data are recorded in Table 1.

## Correlation tests
### *Pearson's product-moment correlation tests*
We performed a series of correlation tests in order to evaluate the extent to which variations in vascular growth proxies were dependent upon size and robusticity, which bear a strong influence on many aspects of organismal biology (*Peters, 1983*; *Calder, 1984*; *Stearns, 1992*). For instance, small-bodied therocephalians might have achieved their increasingly diminutive sizes by having a slower growing, less vascularized skeleton compared to their larger-bodied predecessors. In this scenario, one would expect a correlation between smaller size and slower growth, and between larger size and faster growth across subclades. A similar correlation in which small-bodied species generally exhibit slower growth rates than larger-bodied groups has been identified in extant vertebrates (*Case, 1978*), and suggested by histologic data in fossil non-avian dinosaurs (*Erickson et al., 2004*; *Lee, 2007*). On the other hand, smaller sizes may have been achieved by shortening the duration of the growth period, in which case no generalized correlations between size and vascular proxies of skeletal growth are necessary. To address these hypotheses, histomorphometric data (%*CV*, *POD*, *RBT*) and midshaft cross-sectional area were recorded for each sectioned limb bone. The data were organized into propodial, epipodial, and pooled subsets to control for the effects of increased variance from pooling limb bones of different types (although the effects appeared to be minimal as all tests ultimately yielded similar results). For each data partition, Pearson's product-moment correlation tests were performed between histomorphometric variables (%*CV*, *POD*, *RBT*) and the natural log of midshaft cross-sectional area. Vascular growth proxies were also tested for correlations with bone robusticity independent of size, and with each other to assess whether %*CV* and *POD* provided comparable estimates of vascularization.

### Phylogeny-independent contrasts

Independent contrast methods were carried out to control for the effects of phylogenetic non-independence of putative correlations (*Felsenstein, 1985*; *Garland, Bennett & Rezende, 2005*). For example, one might find low tissue vascularity in Triassic baurioids due to their generally small sizes, or due to their close relatedness (and, by extension, their inherited phenotypic similarities). We carried out additional correlation tests on an augmented data set using the PDAP:PDTREE module (*Midford, Garland & Maddison, 2011*) in Mesquite version 2.0 (*Maddison & Maddison, 2007*). This required a tree and branch lengths (adapted from *Huttenlocker, 2013*; *Huttenlocker, 2014*) pruned to the 11 histologically sampled taxa. First, tip data for each of the 11 taxa were recorded in a NEXUS file, including average *%CV*, *POD*, *RBT* and the natural log of midshaft area of the propodials and epipodials separately (limb bones were not pooled for independent contrasts). Second, ancestral character states (estimated using squared-change parsimony) were checked for the assumption of Brownian motion evolution that governs independent contrasts (*Felsenstein, 1985*; *Díaz-Uriarte & Garland, 1996*). This was performed by running assumption-testing operations in PDAP that evaluate the relationship between the absolute values of independent contrasts and their corresponding standard deviations. Non-significant relationships were determined for each of the data partitions, indicating that a Brownian motion model adequately fit the tip data. Finally, the same sets of regressions were performed on independent contrasts as in the untransformed data (size, robusticity, and vascular growth proxies).

## RESULTS

### Results 1: qualitative histomorphology

Due to space limitations, the present description is accompanied by a more detailed account in the Supplemental Data file. In general, limb bone cortices in studied therocephalians were thick and formed primarily by FLB with varying degrees of vascularization, depending on size and phylogenetic position. For example, large basal therocephalians from the Middle Permian exhibited subplexiform FLB in the propodial elements with a comparatively high degree of vascularization (*%CV* ~13–16%; *POD* ~75–168 μm). By contrast, some smaller-bodied taxa in both the Permian and Triassic tended to incorporate more parallel-fibered bone or showed evidence of increased parallel-fibered and lamellar bone deposition with less vasculature toward the outer cortex, indicative of growth attenuation and attainment of somatic maturity (*de Margerie, Cubo & Castanet, 2002*). This pattern is marked in *Mirotenthes*, which preserves an external fundamental system ('EFS') in some elements (an outer collar of avascular, generally acellular, lamellar bone indicative of virtual cessation of circumferential growth in adult individuals; *Cormack (1987)*; *Huttenlocker, Woodward & Hall (2013)*). Systematic cortical remodeling by Haversian bone formation was rare and only a few sparse secondary osteons were identified in some Permian scylacosaurian taxa [e.g., an indeterminate scylacosaurid and *Moschorhinus*, contrary to reports of extensive skeletal remodeling by *Chinsamy-Turan & Ray (2012)*: p. 203)]. As in dicynodonts and other therapsids in which histology is

well known, most basal therocephalians and eutherocephalians exhibited remarkably thick bone walls (∼20–35% *RBT* in forelimb elements of lycosuchids, scylacosaurids, and akidnognathids; ∼40% in forelimbs of hofmeyriids). This condition, however, was lost in baurioids, which were instead characterized by a much thinner cortical bone wall (<25% in the forelimb and propodials) and more slender limb bones.

Permian basal therocephalians and eutherocephalians (Figs. 2A–2C, 2F, 2G; 3A, 3B) generally showed multiple growth zones as well, demarcated by cyclic growth marks indicative of periodic pauses in growth as in other Permian therapsids (*Ray, Botha & Chinsamy, 2004*; *Ray, Botha-Brink & Chinsamy-Turan, 2012*; *Botha-Brink & Angielczyk, 2010*; *Botha-Brink, Abdala & Chinsamy, 2012*). Although Triassic therocephalians (Figs. 2D, 2E; 3C–3F) showed a variety of vascular motifs and tissue-types (e.g., richly vascularized and disorganized bone tissues in akidnognathids; less vascularized parallel-fibered bone tissue in baurioids), fewer growth marks were recorded in all Triassic species. Small Triassic baurioids in particular showed thinner bone walls with few growth marks, less vascularized limb bone cortices, and smaller, more sparsely distributed primary osteons on average.

## Result 2: quantitative histometrics

### Pearson's correlation and phylogeny-independent contrasts

Results of raw and phylogeny-corrected correlation tests are presented in Table 2. Correlation tests on raw data found that vascular growth proxies were strongly positively correlated with size [i.e., the natural log of midshaft cross-sectional area, underscoring greater overall tissue vascularity in large bones from larger taxa (e.g., *Lycosuchus*, scylacosaurids, *Moschorhinus*)] (Fig. 4; Table 2). Both %*CV* and *POD* also exhibited a strong positive correlation with each other, indicating that %*CV* and *POD* represent equivalent proxies for understanding relationships between bone tissue vascularization and growth across the therocephalian clade. Whereas the selected vascular growth proxies shared a consistent positive relationship with size for all data partitions, putative associations between robusticity and size were less clear. Correlation tests on raw size data and *RBT* were non-significant. Likewise, correlation tests on raw vascular growth proxies and *RBT* either yielded non-significant *p*-values or, in the case of the propodial-only data partition, had low correlation coefficients with only a weakly positive association. This result is likely due to the fact that even some small-bodied taxa exhibited unexpectedly thick bone walls, as in the case of the hofmeyriids *Hofmeyria* and *Mirotenthes*. Bone wall thickness correlates poorly with size and growth proxies, suggesting that bone robusticity is not necessarily tied directly to growth (perhaps being constrained by ecology, habitat, or mechanical regimen in different groups of therocephalians).

Results of phylogeny-corrected correlations further indicated a strong relationship between size and some histometrics (but not with limb bone robusticity) (Table 2). Evolutionary decreases in cross-sectional area of limb bones were generally associated with decreases in average %*CV* and *POD*. Eutherocephalians and particularly baurioids demonstrated a noteworthy pattern in which reconstructed ancestor-descendant size

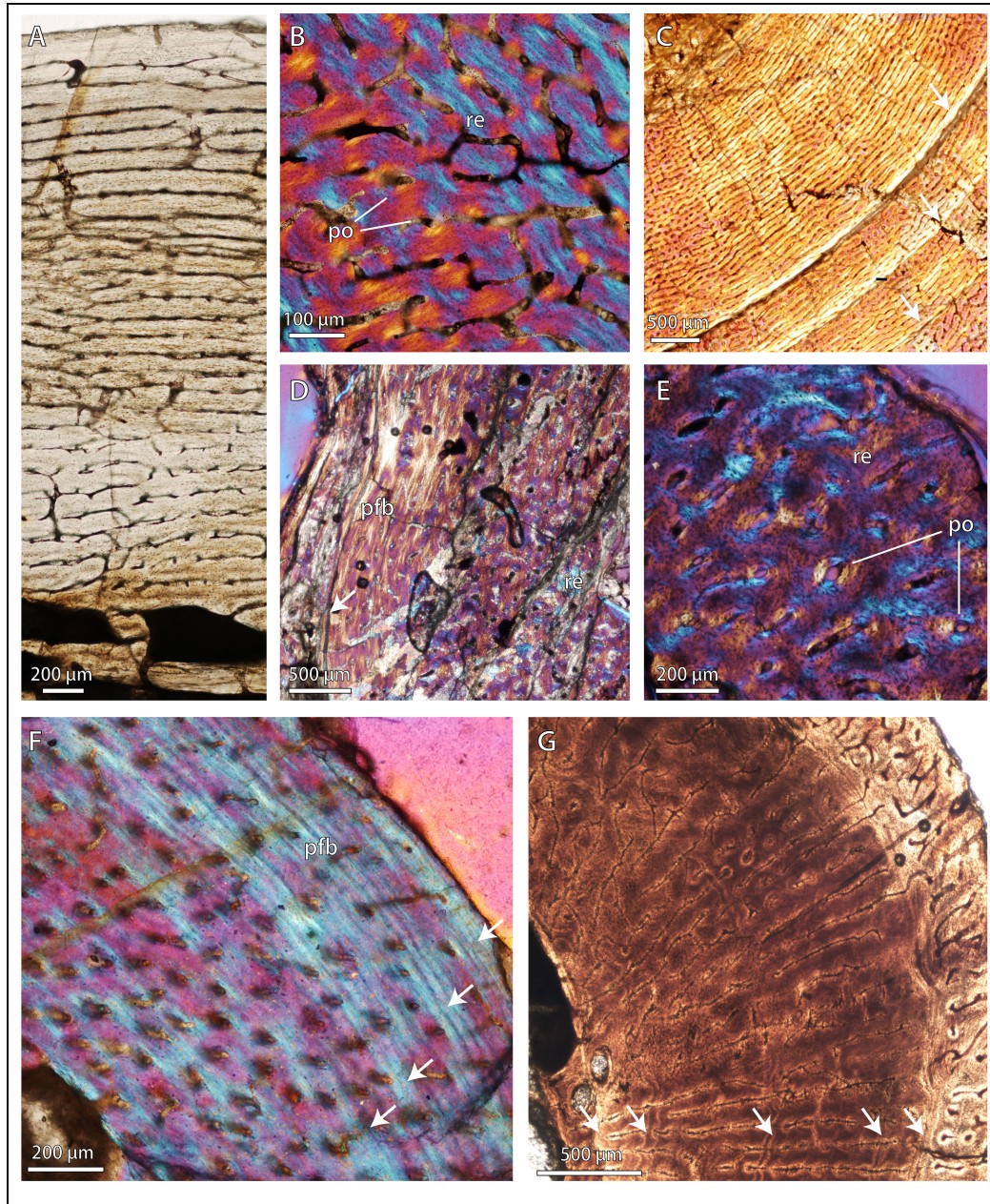

**Figure 2 Bone microstructure in selected basal therocephalians and Permo-Triassic euthero-cephalians.** (A) Middle Permian *Lycosuchus* (SAM-PK-K9012), femur midshaft, dorsal cortex showing subplexiform fibrolamellar bone viewed under non-polarized light. (B) Middle Permian *Lycosuchus* (SAM-PK-9084), ulna midshaft, cortical fibrolamellar bone viewed at high magnification showing reticular canals in a woven-fibered matrix (crossed-nicols with wave plate). (C) Scylacosauridae indet. (CGS R300), humerus midshaft, cortex viewed at low magnification showing growth marks (crossed-nicols with wave plate). (D) Early Triassic akidnognathid *Olivierosuchus* (NMQR 3605), humerus midshaft, cortical fibrolamellar bone showing thick zone of reticular fibrolamellar bone followed by parallel-fibered bone and a LAG (crossed-nicols with wave plate). (E) *Olivierosuchus* (SAM-PK-K10617), femur midshaft, close-up of primary osteons and woven-fibered interstitial matrix (crossed-nicols with wave plate). 

**Table 2** Pearson's product-moment correlation statistics (Pearson's *r* and *p*) for size, robusticity, and vascular growth proxies in therocephalians.

| | Propodial-only | | Epipodial-only | | Pooled | |
|---|---|---|---|---|---|---|
| | *r* | *p*[*] | *r* | *p* | *r* | *p* |
| **All therocephalians, raw data** | | | | | | |
| ln midshaft area vs. *RBT* | 0.309 | 0.109 | −0.186 | 0.267 | −0.126 | 0.301 |
| ln midshaft area vs. %*CV* | **0.744** | **<0.001** | **0.714** | **<0.001** | **0.746** | **<0.001** |
| ln midshaft area vs. *POD* | **0.853** | **<0.001** | **0.847** | **<0.001** | **0.850** | **<0.001** |
| *RBT* vs. %*CV* | **0.452** | **0.012** | 0.034 | 0.839 | 0.149 | 0.217 |
| *RBT* vs. *POD* | **0.360** | **0.050** | 0.111 | 0.512 | 0.143 | 0.237 |
| %*CV* vs. *POD* | **0.760** | **<0.001** | **0.802** | **<0.001** | **0.796** | **<0.001** |
| **All therocephalians, independent contrasts** | | | | | | |
| ln midshaft area vs. *RBT* | −0.466 | 0.174 | −0.476 | 0.232 | – | – |
| ln midshaft area vs. %*CV* | **0.765** | **0.006** | **0.807** | **0.015** | – | – |
| ln midshaft area vs. *POD* | **0.656** | **0.028** | **0.859** | **0.006** | – | – |
| *RBT* vs. %*CV* | −0.077 | 0.831 | −0.224 | 0.592 | – | – |
| *RBT* vs. *POD* | −0.472 | 0.168 | −0.010 | 0.979 | – | – |
| %*CV* vs. *POD* | **0.639** | **0.034** | **0.877** | **0.004** | – | – |

**Notes.**

[*] *r* and *p*-values in boldface are significant at $\alpha = 0.05$.

reductions of Late Permian and Triassic lineages were associated with decreases in the average level of tissue vascularization. Patterns in limb bone robusticity were less clear. As in the analyses of the raw data, phylogeny-corrected correlations between *RBT* and size or growth proxies were non-significant.

# DISCUSSION

## General histological patterns

### Size, robusticity, and vascular growth proxies

Biologically meaningful associations are detectable in therocephalian histology, as in some other therapsid groups. For example, vascular growth proxies were strongly positively correlated with size as larger-bodied species typically exhibited greater overall tissue vascularity (e.g., *Lycosuchus*, scylacosaurids, *Moschorhinus*). Previous tests incorporating a large histologic sample of dicynodonts and other therapsids found similar correlations between raw vascular growth proxies and size (estimated from skull lengths), although

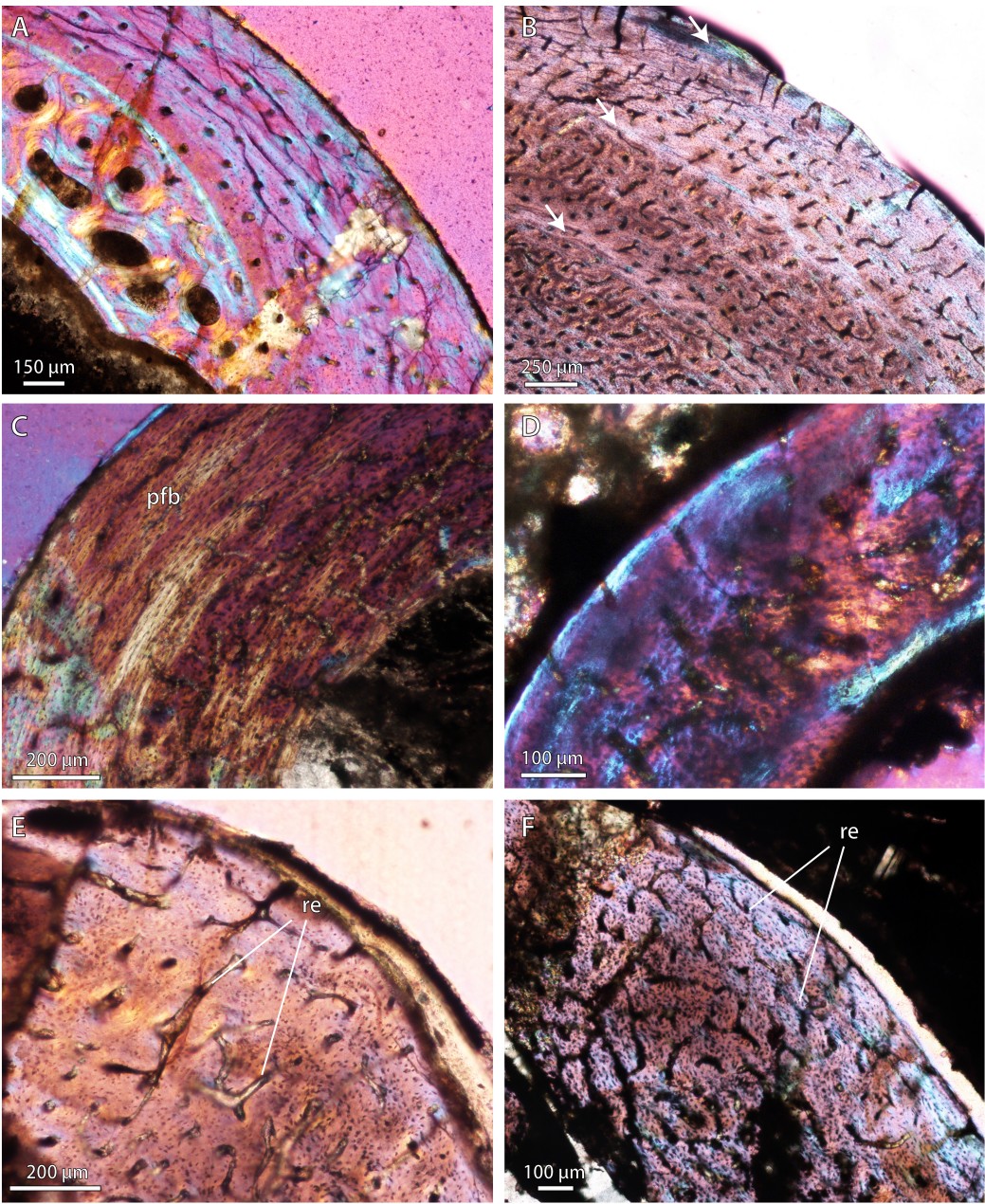

**Figure 3 Bone microstructure in selected Permo-Triassic baurioid eutherocephalians.** (A) Late Permian *Ictidosuchoides* subadult (SAM-PK-8659), radius midshaft, cortex and inner cancellous bony scaffold (crossed-nicols with wave plate). (B) *Ictidosuchoides* adult (BP/1/4092), humerus midshaft, cortex with longitudinal and reticular primary osteons and growth marks (crossed-nicols with wave plate). (C) Early Triassic *Tetracynodon* (NMQR 3745), humerus midshaft, cortex showing inner fibrolamellar bone and sparsely vascularized outer parallel-fibered bone (crossed-nicols with wave plate). (D) Early Triassic *Scaloposaurus* (SAM-PK-K4638), humerus midshaft, cortex showing longitudinal and reticular fibrolamellar bone deposition followed by a thin collar of lamellar bone in the subperiosteal region (crossed-nicols with wave plate).

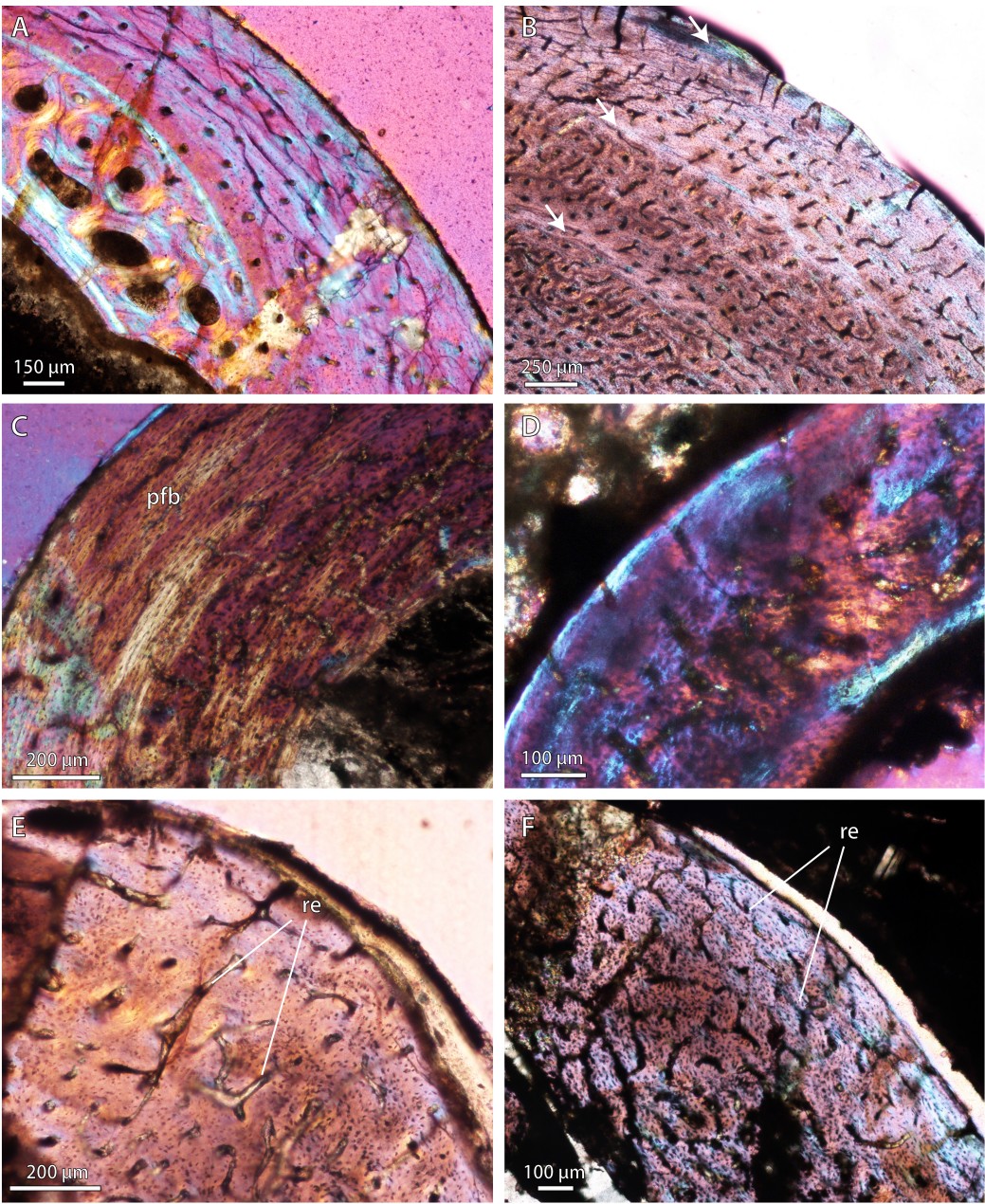

**Figure 3 (...continued)**

(E) Middle Triassic bauriid *Microgomphodon* (NMQR 3605), humerus midshaft, cortex showing reticular fibrolamellar bone with no growth marks (crossed-nicols with wave plate). (F) *Microgomphodon* (NMQR 3605), femur midshaft, cortex showing reticular fibrolamellar bone (crossed-nicols with wave plate). Arrows denote growth marks. Detailed descriptions and photomicrographs are provided in the Supplemental Data file. Abbreviations: pfb, parallel-fibered bone; re, reticular fibrolamellar bone.

tests on a subset of dicynodonts were only marginally significant and non-significant when independent contrasts were evaluated (*Botha-Brink & Angielczyk, 2010*). The new data agree with these earlier results by large measure. However, the present analysis found a strongly positive correlation between size and both vascular growth proxies (%*CV* and *POD*) even when corrected for phylogeny. Prior analyses implementing phylogeny-independent contrasts on dicynodonts were unable to identify similar patterns, despite a significant correlation between the raw data (*Botha-Brink & Angielczyk, 2010*). Improved statistical results in this study are a likely consequence of sampling bones of particular types separately (rather than averaging vascularity across all bones) and using cross-sectional measurements from sampled bones as a size proxy (rather than representative skull lengths from specimens that were not sampled histologically). Tests on raw cortical thickness (*RBT*) in dicynodonts also showed no clear association with size or degree of vascularization as in the present study, even though a positive association with size was discovered when corrected for phylogeny. No correlation was observed between bone robusticity (*RBT*) and size or degree of vascularization in therocephalians (raw or phylogeny-corrected), suggesting that size and rate of growth may have had limited influence over bone robusticity as compared to other aspects of organismal biology such as mechanical regime (e.g., locomotor behavior, stance or gait; *Currey & Alexander, 1985*) and ecology or habitat preference (e.g., burrowing, semi-aquatic/aquatic; *Wall, 1983*; *Laurin, Girondot & Loth, 2004*; *Germain & Laurin, 2005*). Similarly, data on the genus *Moschorhinus* indicated that overall bone compactness was related to the thickness of the bone wall, but was largely independent of growth and degree of vascularity (*Huttenlocker & Botha-Brink, 2013*).

## Phylogenetic patterns

Previous interpretations of growth patterns in early therocephalians were based on limited information from incomplete specimens (*Ricqlès, 1969*; *Ray, Botha & Chinsamy, 2004*; *Chinsamy-Turan & Ray, 2012*). Additional specimens described here suggest that at least some large-bodied predators from the Middle Permian, including *Lycosuchus* and some scylacosaurids, exhibited subplexiform FLB in propodial elements. Some of the larger-bodied predators in the sample also showed the highest degree of cortical vascularization. This is remarkable, as subplexiform FLB is one of the most rapidly deposited tissue-types in archosaurian and mammalian limb bones, and similarly vascularized bone in birds and mammals forms periosteally at a rate generally greater than 15 µm/day (*Castanet et al., 2000*; *de Margerie, Cubo & Castanet, 2002*; *Cubo et al., 2012*). It is noteworthy that the subplexiform tissue complex, present in *Lycosuchus* and scylacosaurids, represents the prototype upon which earlier workers first described FLB [see *Stein & Prondvai (2013)*

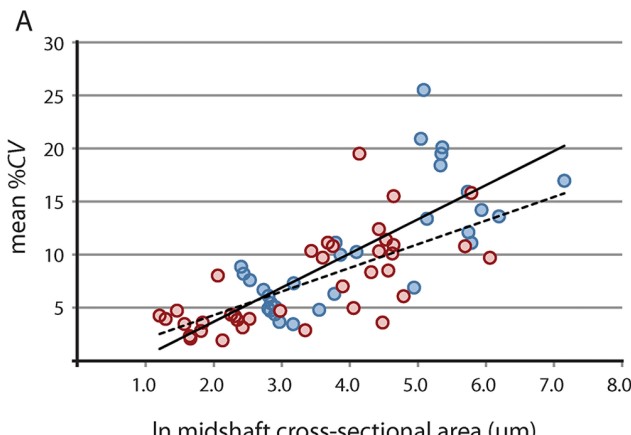

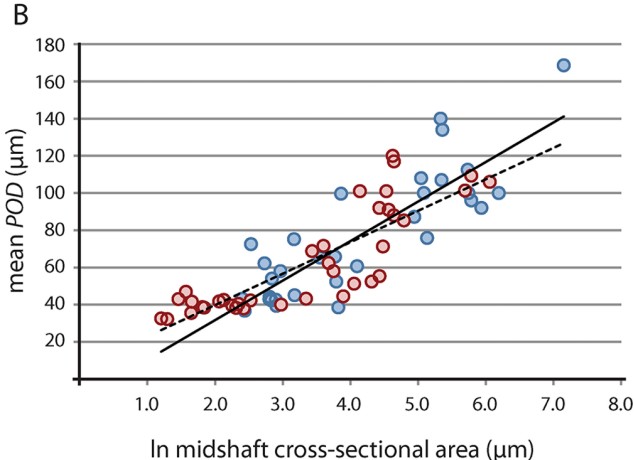

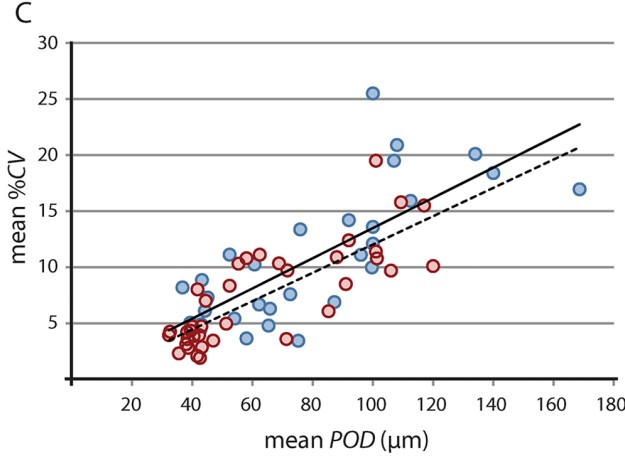

**Figure 4 Linear regression of vascular growth proxies against size (midshaft cross-sectional area) from limb bone elements.** (A) Mean cortical vascularity (%CV) against midshaft cross-sectional area. (B) Mean primary osteon diameter (POD) against midshaft cross-sectional area. (C) mean %CV against mean POD. All correlations between vascular growth proxies and size are strongly positively correlated for both propodials (solid regression line) and epipodials (dashed regression line) (see statistical results in Table 2). Blue circles, propodials. Red circles, epipodials.

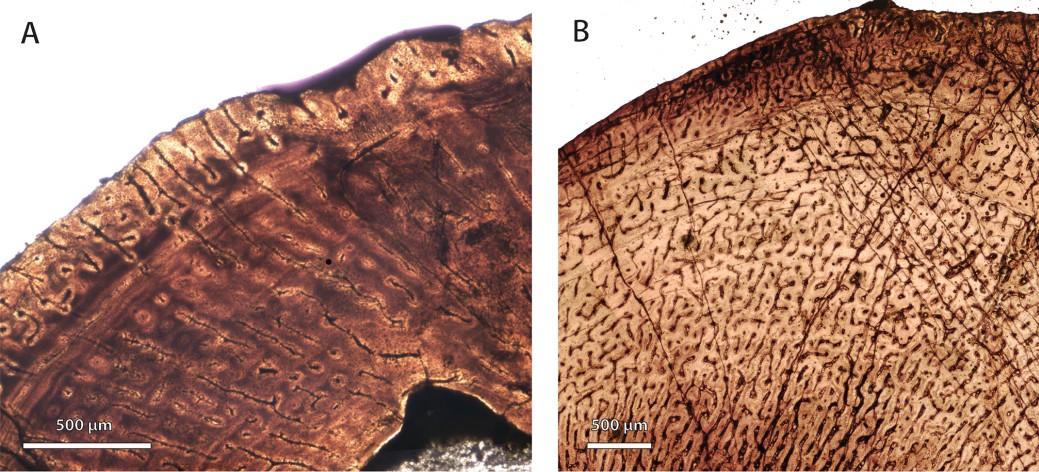

**Figure 5 Comparison of bone histology and microvasculature in Permian *Theriognathus* and *Moschorhinus*.** (A) *Theriognathus* (NMQR 3375) femur midshaft cortex (non-polarized light). (B) *Moschorhinus* (NMQR 3939) humerus midshaft cortex (non-polarized light). Note the greater overall degree of vascularity in 'B.'

for a review]. This subplexiform condition is conspicuously lacking in the whaitsiid *Theriognathus* (Fig. 5) and some later eutherocephalians. Consequently, the evolutionary scenario suggested by *Ricqlès (1969)* and critically re-examined by *Chinsamy-Turan & Ray (2012)* should be revised: basal therocephalians exhibited highly vascularized FLB and grew relatively rapidly over many growing seasons but with frequent interuptions, in contrast to some later eutherocephalians (with the exception of the akidnognathid *Moschorhinus*, which maintained relatively fast growth).

Moreover, elevated vascularity and rapid growth may be a characteristic of larger bodied taxa, an interpretation that is supported by observations in other large predatory theriodonts such as the cynodont *Cynognathus* and some large gorgonopsians (*Botha-Brink, Abdala & Chinsamy, 2012*; *Chinsamy-Turan & Ray, 2012*). Importantly, evolutionary decreases in body size during the Permian (*Huttenlocker, 2013*; *Huttenlocker, 2014*) were associated with decreases in overall degree of skeletal vascularization. Both hofmeyriids and *Theriognathus* (in spite of the latter's substantial size) showed modest cortical vascularity and smaller mean primary osteon diameters than in basal lycosuchids, scylacosaurids or akidnognathids, but the baurioids (which are deeply nested in the tree and generally smaller-bodied) had the least vascularized bone tissues (Fig. 6). These clade-level patterns suggest that reductions in body size of some Late Permian eutherocephalians were coupled with decreased rates of skeletal apposition leading up to the end-Permian extinction, particularly in the baurioid lineage.

## Growth and the end-permian extinction

The end-Permian extinction was associated with evidence of temporary body size reductions, restricted primarily to faunas of the earliest Triassic (Induan) (although size reductions in some therapsids may reflect larger-scale patterns of body size evolution

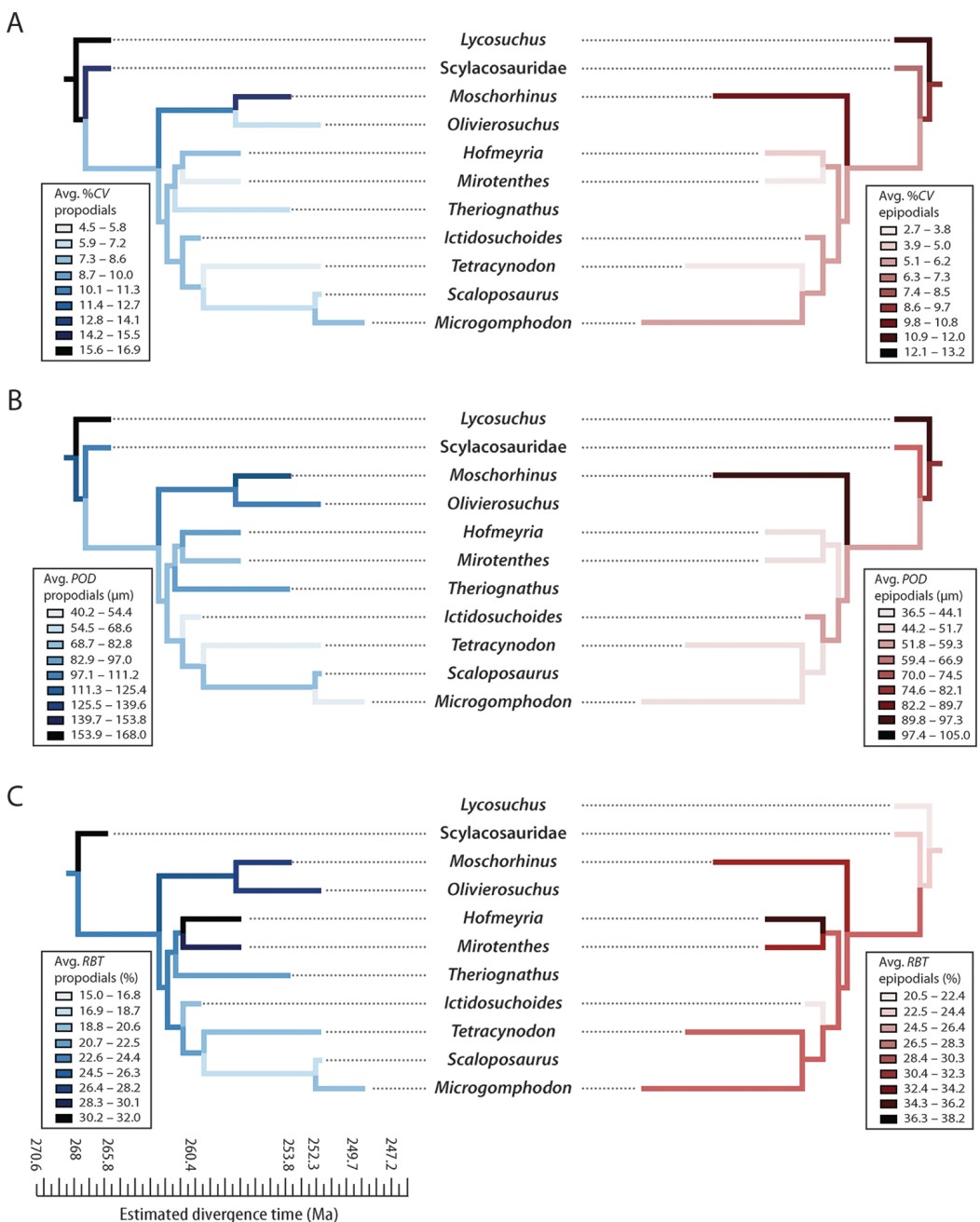

**Figure 6 Mirror phylogenies of Permo-Triassic therocephalians sampled for bone histology (scaled to geologic time).** Phylogenetic character mapping of histological traits estimated from propodials (left) and epidpodials (right) reveals comparable ancestor-descendant changes for each pool of skeletal elements. (A) Average cortical vascularity (%*CV*). (B) Average primary osteon diameter (*POD*). (C) Relative bone wall thickness (*RBT*). Ancestral states were reconstructed using squared-change parsimony in Mesquite version 2.0 (*Maddison & Maddison, 2007*).

tracing back to the Permian; *Huttenlocker, 2013*; *Huttenlocker, 2014*). Such size shifts have been documented previously based on invertebrate burrows, foraminifera, brachiopods, gastropods, bivalves, conodonts, and fish (*Twitchett & Barras, 2004*; *Payne, 2005*; *Twitchett, 2007*; *Luo et al., 2008*; *Mutter & Neuman, 2009*; *Metcalfe, Twitchett & Price-Lloyd, 2011*; *Song, Tong & Chen, 2011*; *Rego et al., 2012*). Nevertheless, questions remain regarding the evolution of growth patterns and their underlying influence on size shifts (*Twitchett, 2007*; *Harries & Knorr, 2009*). For example, organisms may have experienced slower overall growth rates in response to more limited resources or environmental degradation during the Permian-Triassic transition. Growth mark analyses on marine brachiopod shells (i.e., '*Lingula*') have indicated slow growth rates with frequent interruptions to growth in earliest Triassic shells (*Metcalfe, Twitchett & Price-Lloyd, 2011*). The authors attributed their results to suboptimal environmental conditions, including episodes of benthic hypoxia, hypercapnia, ocean acidification, and/or disruptions to primary productivity. Such environmental factors would place strong physiological limits on shell formation. However, identifications of lingulid specimens were tenuous given conservatism in external morphology of the group, and growth marks were only studied in shells from Triassic survival and recovery faunas without being compared to Permian specimens. Alternatively, surviving lineages could have exhibited heterochronic shifts shortening time to maturity (e.g., progenesis) and thus rapid, sustained growth over a brief growth period. Progenesis is a classic example of an *r*-selection strategy in perturbed or unstable environments (*Gould, 1977*) and has been identified as a potential mechanism of some Lilliput patterns (*Harries, Kauffman & Hansen, 1996*). Finally, if shifts in body size distributions were influenced primarily by differential extinction of large- versus small-bodied forms, then there may have been no resulting changes in growth patterns at all (that is to say, changes in growth dynamics are not necessary to explain post-extinction body size distributions).

### Growth patterns in Permo-Triassic therapsids compared

New data on growth patterns in nonmammalian therapsids, as well as other Permo-Triassic tetrapods, offer the potential to further evaluate patterns of selectivity during mass extinctions (*Botha-Brink & Angielczyk, 2010*; *Botha-Brink & Smith, 2012*). Bone histology has been studied in numerous genera of Permian and Triassic dicynodont therapsids (*Chinsamy & Rubidge, 1993*; *Ray, Botha & Chinsamy, 2004*; *Ray, Bandyopadhyay & Bhawal, 2009*; *Ray, Botha-Brink & Chinsamy-Turan, 2012*) and recent progress in dicynodont paleobiology has permitted evolutionary investigations of their growth patterns (*Botha-Brink & Angielczyk, 2010*). Increased rates of skeletal growth originated relatively early, either within or prior to the divergence of bidentalian dicynodonts (e.g., *Dicynodon*, *Lystrosaurus*) by the early-Late Permian. Medium-to-large Late Permian dicynodonts continued to show patterns of increased cortical vascularity, especially within the Permo-Triassic boundary-crossing genus *Lystrosaurus*, which demonstrated some of the highest levels of tissue vascularity (~20%). Triassic specimens of *Lystrosaurus* showed relatively higher vascularity and fewer growth marks. In the therocephalian *Moschorhinus*, the only large therapsid predator to cross the Permian-Triassic boundary, within-lineage

size reduction was associated with the maintenance of rapid but attenuating growth over a short period (*Huttenlocker & Botha-Brink, 2013*). This pattern suggests that *Moschorhinus* may have been under selection for reaching its adult size relatively more quickly than in other Permian therocephalians. Notably, *Moschorhinus* exhibited comparably high levels of tissue vascularity as *Lystrosaurus* during the Triassic (∼20–25%).

Whereas anecdotal evidence in *Lystrosaurus* and *Moschorhinus* is suggestive of within-lineage heterochronic shifts, clade-level patterns introduce a more complex explanation for observed body size reductions in therocephalians. In particular, body size reductions occurred early during the evolution of eutherocephalians and were associated with a lesser degree of cortical vascularity in medium-to-small-bodied Permian and Triassic forms (e.g., hofmeyriids and especially baurioids). The two major subclades of therocephalians that persisted into the earliest Triassic, Akidnognathidae and Baurioidea, revealed distinctly different growth patterns from each other as interpreted through their tissue texture and degree of vascularity. Their histology suggests a bimodality of life history strategies in earliest Triassic therocephalians: small-to-medium akidnognathids with well-vascularized (fast-growing) bone and smaller-bodied baurioids with less vascularized (slower-growing) bone. However, both groups shared a reduced number of growth marks compared to their Permian relatives in addition to their generally smaller sizes. This nuance is not evident from the quantitative analysis based on vascular proxies of growth rate alone, but is evident from growth mark counts in surveyed specimens. Permian theriodonts that have been sampled histologically, including some gorgonopsians, the cynodont *Procynosuchus*, basal therocephalians, Permian (but not Triassic) specimens of *Moschorhinus*, hofmeyriids, and *Theriognathus*, typically showed evidence of prolonged, multi-year growth often to larger body sizes, a pattern that is not represented in earliest Triassic therapsids sampled to date (*Botha-Brink & Angielczyk, 2010*; *Botha-Brink, Abdala & Chinsamy, 2012*).

The present discussion of associations between micro- and macrostructure provides a more functional and wholistic context for the evolution of histological features in Permian and Triassic therocephalians. However, it is important to note that the patterns discussed here have not been evaluated quantitatively and therefore merit future investigation. Recent large-scale phylogenetic studies in other therapsids have addressed how life history and functional morphology might have contributed to the success of some groups during the Late Permian and Triassic (e.g., bidentalian dicynodonts). For example, the degree of development of the secondary palate has been linked anecdotally to new environmental conditions with the onset of the Early Triassic, particularly a dramatic decline in atmospheric $pO_2$ from the Late Permian and continuing through the Middle and Late Triassic (*Retallack, 2003*). However, a rigorous collections-based study found no difference between secondary palate length in Permian and Triassic dicynodonts when corrected for size and phylogeny (*Angielczyk & Walsh, 2008*). Similarly, a bony secondary palate can be found in a number of therocephalian subgroups, both in the Permian and Triassic, and Triassic therocephalian faunas consisted of species both with (e.g., baurioids) and without (e.g., akidnognathids) a secondary palate (Fig. 7). No clear association with

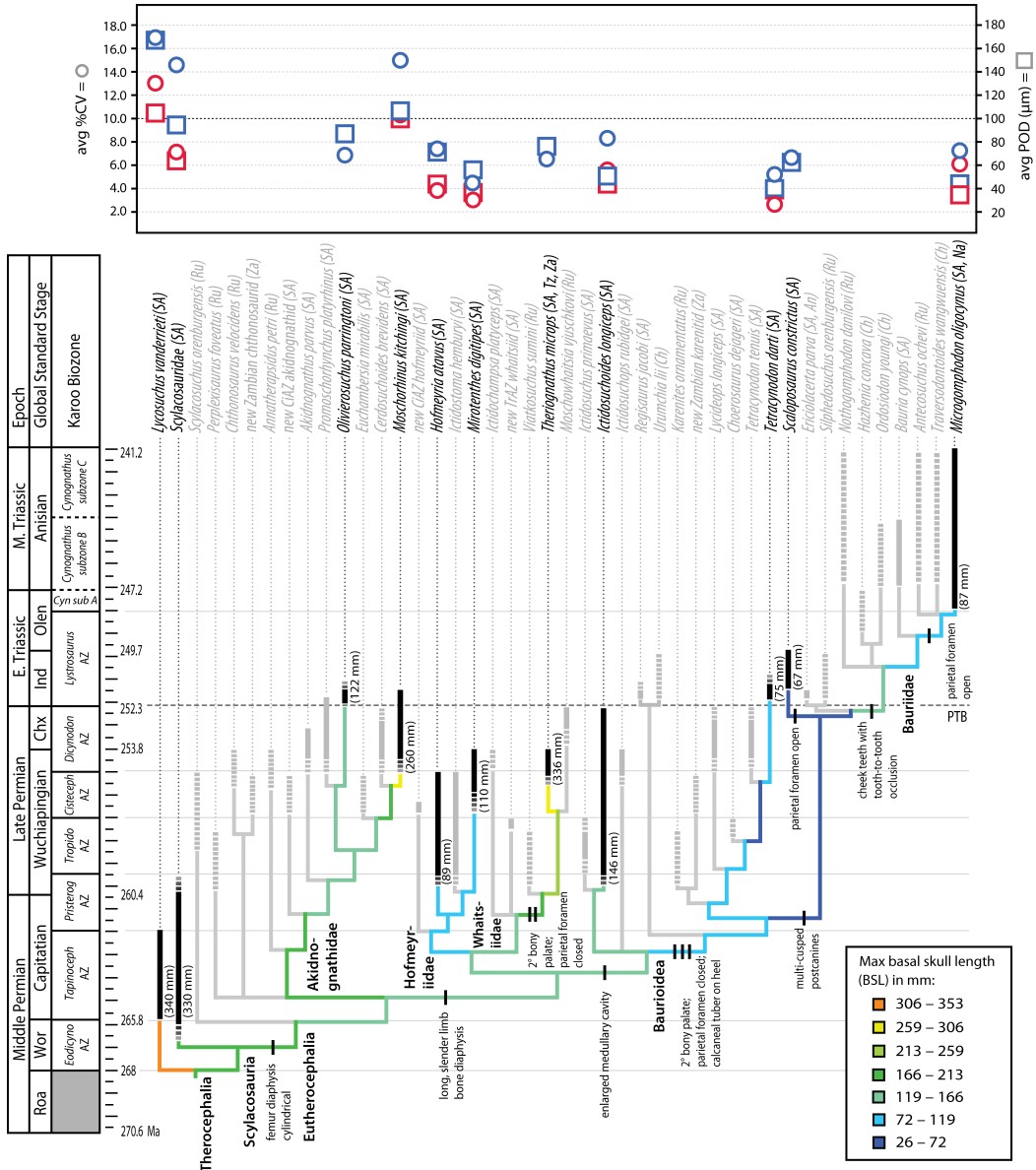

**Figure 7 Summary of evolution of size and bone microstructural traits.** Black bars represent stratigraphic ranges of taxa that were sampled histologically. Numbers in parentheses to right of black bars denote maximum size of taxon in interval of first appearance. Tree topology and ranges from Huttenlocker, 2013 and 2014. In graph at top of figure, circles represent average %*CV* and squares average *POD* of propodials (blue) and epipodials (red). Abbreviations: Chx, Changxingian; *Cisteceph* AZ, *Cistecephalus* Assemblage Zone; *Cyn* sub A, *Cynognathus* subzone A; *Eodicyno* AZ, *Eodicynodon* Assemblage Zone; Ind, Induan; Olen, Olenekian; *Pristerog* AZ, *Pristerognathus* Assemblage Zone; Roa, Roadian; *Tapinoceph* AZ, *Tapinocephalus* Assemblage Zone; *Tropido* AZ, *Tropidostoma* Assemblage Zone; Wor, Wordian.

global hypoxia and respiratory efficiency can be made based on this character, and, as a result, other factors may have been more important in maintaining and shaping the evolution of the secondary palate (e.g., feeding mechanics; *Thomason & Russell, 1986*).

Similar studies addressing the possible effects of hypoxia on cortical tissue vascularity have found few differences between Permian and Triassic therapsids, instead demonstrating that highly vascularized tissues with enlarged canals evolved early in bidentalian dicynodonts (*Botha-Brink & Angielczyk, 2010*). In therocephalians, variation in the degree of vascularization of limb cortices is best explained by size variation observed across clades (Fig. 7). Most smaller-bodied groups exhibited less vascularized limb bone cortices, a character that evolved relatively early in the evolutionary history of eutherocephalians and held over in some small Triassic taxa. The smallest Triassic forms, derived bauroids, typically had lighter, more gracile skeletons with an open medullary cavity, thinner bone walls, and few to no growth marks. In addition to longer, more slender limb bones, they also had an elongate hind foot with a calcaneal tuber on the heel, maxillary bridge forming a bony secondary palate (discussed above), increasingly specialized multi-cusped teeth, and often lacked a parietal foramen (or reduced the pineal body altogether as in *Tetracynodon*; *Sigurdsen et al., 2012*). The selective value of maintaining a parietal eye for temperature regulation and modulating melatonin production may have been diminished in some small, nocturnal or crepuscular baurioid therocephalians, or in short-lived animals less dependent on seasonal cues in photoperiodicity (*Roth, Roth & Hotton, 1986*). The latter scenario is consistent with the lack of cyclic bone deposition and paucity of growth marks in small Triassic bauroids (although one genus, *Scaloposaurus*, is distinguished from other small Triassic bauroids in its retention of the parietal foramen and pineal body).

## CONCLUSION

A survey of histological patterns in therocephalians found that limb bone cortices composed of thick deposits of FLB with cyclic growth marks were widespread in early therocephalians, but evolutionary decreases in adult body sizes of some clades were associated with reductions in cortical vascularity and skeletal growth leading up to the end-Permian mass extinction. In Permo-Triassic therocephalians, a pattern of multi-year growth to large body size that was common in the Permian was selected against in the earliest Triassic. This conclusion is supported by (1) ecological removal of large-bodied taxa having prolonged, multi-year growth patterns; (2) cladistically inferred survival of small-bodied taxa with modest skeletal apposition rates, but truncated growth durations (i.e., bauroids); and (3) within-lineage shifts in growth patterns observed in boundary-crossing genera in the Karoo (i.e., *Lystrosaurus* and *Moschorhinus*). A synergistic combination of local within-lineage effects and differential extinction patterns strongly influenced Triassic Lilliput faunas, weakening the hypothesized role of rapid adaptive evolution of new small-bodied forms. Similar within-lineage size decreases and size selective extinctions contributed strongly to Lilliput patterns in marine gastropods, foraminifera, and brachiopods, although all three mechanisms have been invoked to

explain Early Triassic foraminifera size distributions globally (*Payne, 2005*; *Metcalfe, Twitchett & Price-Lloyd, 2011*; *Song, Tong & Chen, 2011*; *Rego et al., 2012*). Contrary to the present results, growth mark analyses on lingulid brachiopods are suggestive of different physical factors influencing skeletal growth in marine benthos (due to slowed, but prolonged shell secretion over many years) (*Metcalfe, Twitchett & Price-Lloyd, 2011*). More skeletochronologic and phylogenetic data are needed, however, to understand the generality of these patterns among Permo-Triassic Lilliput taxa in marine and terrestrial realms.

Although some effects on size and growth are observable in Triassic therocephalians (and perhaps therapsids more generally), it is important to note that much of the diversity observed in the earliest Triassic *Lystrosaurus* Assemblage Zone of the Karoo originated during the Late Permian, and that variation in body sizes and growth patterns during the Late Permian was supplanted by increased bimodality in the Early Triassic *Lystrosaurus* AZ: small-to-medium, fast-growing akidnognathids and the still smaller, slower-growing baurioids. Small size and short growth duration were dominant life history strategies of Early Triassic therocephalians in the Karoo, and reductions in size were typically associated with lesser tissue vascularity and growth rates across but not within taxa. Furthermore, although these strategies were apparently common in the post-extinction environment, their success was temporary. Low rates of origination in therocephalians during this time, coupled with small sizes and reduced niche occupation may have afforded the opportunity for more marginalized groups to diversify (e.g., cynodonts and archosauromorphs). Future applications of phylogenetic comparative methods to studies of body size and growth during the Permo-Triassic will enhance our understanding of interplay between macroevolution and extinctions, and will identify areas of phylogeny that correspond to shifts in trait evolution that conferred success on lineages.

## ACKNOWLEDGEMENTS

AKH thanks S Herring, E Nesbitt, C Sidor, and G Wilson for providing helpful suggestions on the project. F Abdala and B Rubidge (BP), E Butler (NMQR), E De Kock and J Neveling (CGS), P Holroyd (UCMP), A Chinsamy-Turan (University of Cape Town) and S Kaal and R Smith (SAM) are acknowledged for specimen access. F Abdala and K Angielczyk offered additional insights into body size evolution and growth, as well as therapsid diversity patterns during the end-Permian extinction. The National Museum preparatory staff also provided assistance and preparation of some specimens for histological processing: J Nyaphuli, J Mohoi, N Ntheri, S Stuurman, S Ledibane, S Chaka, and W Molehe. Digital image processing and some histologic preparation performed by J Lungmus (University of Washington). F Abdala, A Crompton, and C Organ are acknowledged for their kind and constructive reviews of this manuscript.

## Appendix 1

**List of institutional abbreviations**

**BP**     Evolutionary Studies Institute (previously Bernard Price Institute for Palaeontological Research), University of Witwatersrand, Johannesburg

**CGS**   Council for Geoscience, Pretoria (former Geological Survey field numbers)

**NMQR**  National Museum, Bloemfontein

**SAM**   Iziko South African Museum, Cape Town

**UCMP**  University of California Museum of Paleontology, Berkeley

### Funding

Funding to AKH was provided by the National Science Foundation Doctoral Dissertation Improvement Grant Program (1209018), Evolving Earth Foundation, University of Washington Department of Biology WRF-Hall Fellowship, Burke Museum of Natural History and Culture Vertebrate Paleontology Fellowship, and the Society of Vertebrate Paleontology Richard Estes Memorial Grant. Funding to JB-B was provided by the National Research Foundation (Early Triassic Recovery Project, NRF-65244), Republic of South Africa. The funders had no role in the study design, data collection and analysis, decision to publish, or preparation of the manuscript.

### Grant Disclosures

The following grant information was disclosed by the authors:
National Science Foundation Doctoral Dissertation Improvement Grant Program: 1209018.
Evolving Earth Foundation.
University of Washington Department of Biology WRF-Hall Fellowship.
Burke Museum of Natural History and Culture Vertebrate Paleontology Fellowship.
The Society of Vertebrate Paleontology Richard Estes Memorial Grant.
National Research Foundation (Early Triassic Recovery Project): NRF-65244.

### Competing Interests

The authors declare there are no competing interests.

### Author Contributions

- Adam K. Huttenlocker conceived and designed the experiments, performed the experiments, analyzed the data, contributed reagents/materials/analysis tools, wrote the paper, prepared figures and/or tables, reviewed drafts of the paper.
- Jennifer Botha-Brink performed the experiments, reviewed drafts of the paper.

### Supplemental Information

Supplemental information for this article can be found online at http://dx.doi.org/10.7717/peerj.325.

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
