# Peer review of "Bone microstructure and the evolution of growth patterns in Permo-Triassic therocephalians (Amniota, Therapsida) of South Africa"

_PeerJ, doi:10.7717/peerj.325_

## Round 0.1 · original submission · Major Revisions

As you can see from the reviews, both reviewers thought the manuscript has a lot of interesting material, but reviewer 1 suggests some additional statistical tests and reviewer 2 suggests some reorganization and rewriting, especially of the introduction. Reviewer 2 suggests toning down on some of the technical jargon to make the paper more accessible, something which I think PeerJ should advocate. Additionally, reviewer 2 felt that there were too many figures. You can always include supplementary material including additional figures so as to make the paper with the main figures more streamlined and easy to follow.

·

Basic reporting

This study makes a systematic analysis of bone microstructure in extinct synapsids (extinct relatives of mammals). The primary goals of the study are to 1) develop a database of histological data and images for Permian and Triassic therocephalians and cynodonts, and 2) address life history evolution using histomorphology. The methods and approach are generally good, the subject matter is very interesting, and the results convincing. This paper represents a substantial amount of work by the authors, and with some minor revisions would be a great contribution to the literature.

Experimental design

Acceptable

Validity of the findings

Major Concerns
1. The author employs both “ordinary” and phylogenetically-controlled statistical models. A wealth of literature on comparative phylogenetic methods over the past 30 years has resoundingly shown that “ordinary” statistical models should not be used when analyzing data across species. Presenting both results will confuse nearly all readers. The mixing of approaches leads to incorrect interpretations of the results. And the danger is that, by using both “ordinary” and phylogenetically-controlled tests, the reader will just pick the result that best fits in with their line of thinking (post hoc reasoning). Results from “ordinary” statistical models need to be removed from the paper.
2. Multiple hypothesis testing: The authors perform multiple statistical tests (table 2) and should account for testing multiple hypotheses, which increase the probability of getting “significant results” simply by chance. Bonferroni correction is simple to employ and would increase confidence in the results.
3. The authors find evidence supporting a Brownian motion model of evolution. Since this is random walk model where variance increase with time, the authors should be cautious about using parsimony reconstructions, which don’t account for branch lengths and worse, whose reconstructions cannot be statistically tested for significance.
4. Many times the authors comment about how histologic traits change with body size (e.g. “…but evolutionary decreases in adult body sizes of some clades were associated with reductions in cortical vascularity and skeletal growth leading up to the end-Permian mass extinction.”). I don’t disagree with this but think that the authors need to statistically test for this, using for example femur length as a proxy for body size. Body size should be included in table 2 (which also needs a key for abbreviations), and figure 14. Without such tests, it is hard to know if the mentioned patterns are real.
Minor Concerns
5. Descriptions: These are numerous and thorough. They represent so much text, I think the paper would read better if they were moved to supplemental material.
6. I would really like to see a discussion about how the growth patterns the authors observe contributed to the evolution of determinant growth in mammals. I think it would broaden the interest of the paper.
7. Figure 12: are these regression models controlling for phylogeny? If not, they need to be. It would be good to include the model and significance in the caption or, better, the figure itself. Also, the data points do not look normally distributed. If they are not, you should transform your data prior to analysis (e.g. log).
8. I am a big fan of mirror tree figures, and think this figure 14 is very effective here. Looking at the distribution of the data on the tree, it looks like it could be undergoing directional change, suggesting a persistent pattern of selection or constraint. You may want to test for directional evolution on the tree (correlation between a trait and total path length to the root). This could be a fun addition to the paper, though I wouldn't hold the paper back from publication for its absence.

Minor Comments
Throughout: consistency “densely packed” or “densely-packed”, both are used
Throughout: consistency “larger bodied” or “larger-bodied”, both are used
Throughout: consistency “obliquely oriented” or “obliquely-oriented”, both are used
Throughout: consistency “slower growing” or “slower-growing”, both are used
Throughout: consistency “thinner walled” or “thinner-walled”, both are used
Throughout: consistency “well concentrated” or “well-concentrated”, both are used
Throughout: consistency “well vascularized” or “well-vascularized”, both are used
P. 2: “…dicynodont Lystrosaurus) and cladistically inferred survival of…” change “cladistically” to “phylogenetically” since the methods employ branch lengths.
P. 2: “…New histologic data and phylogenetic comparative methods…” makes it sound as if new comparative methods were developed here.
P. 5: “…suggesting to workers that..” change to “…suggesting that…”
P. 8: add reference ImageJ
P. 26: “…lamaller bone with sparse vascular canals.”; should be “lamellar”
P. 38: “…of longitunally oriented trabeculae...”; should be “longitudinally”
P. 47: “…rates with frequent interuptions to growth in earliest…”; should be “interruptions”

·

Basic reporting

Before accepting this paper I suggest a fair amount of revision is necessary.
The introduction lacks focus and the main point of the paper is lost in a confusing account of bone histology. In recent years the literature has been flooded with accounts of limb mid-shaft histological sections which unfortunately do little to address the biological significance of different bone types in a single bone or in different bones in the same individual.
It would be helpful if the authors gave in their Introduction : 1) the bone types they recognize and how they were formed (given the plethora of terms in use this would be a great help in following this paper) :2) a succinct account of the main features and diversity and history of therocephalians: 2) explain why they have chosen cortical vascularity and osteonal diameter to determine growth rate and the program they used to determine vascularity or at least a reference to the method: 3) Include a brief section contrasting how different growth patterns can be deduced from the histological sections of large and small extant mammals and birds. This would provide a rational for their findings of major differences in bone in two groups of Triassic therocephalians: 4) avoid using abbreviations such as EFS without saying what it is and why it enters into the case they are making.

Experimental design

Not applicable

Validity of the findings

Numerous plates are included in this manuscript, but it is difficult and in many cases impossible to recognize in them the features described in the text. This is especially true for osteonal orientation. The number of bone types illustrated and commented on in this manuscript are overwhelming . Obviously analysis of the biological significance of the different types is beyond the scope of a single paper. However it would help if the authors chose the same bone of the genera selected and described how these document stages of the growth process. This would avoid having to explain why some the bones included in the paper and were subjected to different strain regimes do not always seem to tell the same story. Statistical analysis based on a smaller narrowly focused data set would be more convincing than the information given in the tables.
I find the absence of any discussion of the significance of a lack of secondary osteones in the sections discussed or the biological age of the specimens and unfortunate omission. In extant genera secondary osteones generally reflect either or both age and stress levels. On the other hand they are often absent in small active extant genera. How does this compare with the findings on therocephalians?
The Discussion needs to be rewritten. It tends to ramble and includes peripheral facts and speculations that detract from the main goal of the paper which is presumably to provide a clear statement on the growth processes in Permian and Triassic therocephalians .

Additional comments

This paper represents a tremendous effort of time and dedication. The authors have amassed a mass of new information, but the potential impact of this paper is limited because it’s main audience will be a small group of “paleo-bone” scientists that are dedicated to simply documenting the microstructure of fossil bone. However the information contained in this paper is potentially of great interest to those working on the biology, ontogeny, mechanical properties etc. of living bone. Hence my plea that papers on fossil bone be comprehensible to a broader audience, that they are based on more than sections in a single plane and include sections from different regions of whole bones. Finally, although speculative, attention be paid to the possible mechanical forces to which the bones in question were subjected.

---

## Round 0.2 · Minor Revisions

Overall your manuscript was reviewed favorably by this last, secondary reviewer. Nonetheless this reviewer had several minor comments that, if addressed adequately, should make your paper acceptable for publication. Thank you for your extensive revisions on the original version and for taking care of these final, minor revisions.

·

Basic reporting

no comments

Experimental design

no comments

Validity of the findings

no comments

Additional comments

This represents an important contribution that provides a fresh view to the knowledge of the paleobiology of therocephalians. The sample provided is certainly impressive with new information in the paleohistology of 11 genera, which provide a new benchmark to interpret the biology of therocephalians. The manuscript is suitable for Peer J and should be accepted after minor modifications. I uploaded the pdf of the manuscript with all the pertinent comments.

---

## Round 0.3 · accepted · Accept

Thank you for addressing the reviewer's comments.